# Anxiety and depression among hypertensive patients during the COVID-19 pandemic: A cross-sectional study from Kathmandu Metropolitan, Nepal

**Dilasha K. C.****, Hari Prasad Kaphle, Durga Shrestha****\*, Nirmala Neupane**

School of Health and Allied Sciences, Pokhara University, Kaski, Nepal

\* durgashrestha10.ds@gmail.com

## Abstract

### Introduction

The outbreak of a global pandemic like COVID-19 has highlighted significant distress around mental health. The burden of mental health issues like anxiety and depression requires evidence-based intervention, especially in low-income settings like Nepal. The study aims to determine the prevalence of anxiety and depression and the factors associated with it among hypertensive patients.

### Materials and methods

The quantitative cross-sectional study design was used for this study. The study was conducted among 374 samples from selected wards of Kathmandu Metropolitan using a convenience sampling technique. Face-to-face interviews were conducted using a structured interview schedule. A Chi-square test was used to identify the statistical significance between dependent and independent variables. Binary logistic regression analysis was performed to determine the factors associated with anxiety and depression.

### Results

The prevalence of anxiety and depression among hypertensive patients during the COVID-19 pandemic was 27.8% and 24.3% respectively. According to the results of bivariate logistic regression analysis, smoking/tobacco consumption, staying in quarantine, positive COVID-19 test result, history of COVID-19 positive in the family, History of death due to COVID-19 in the family, visiting a hospital during the COVID-19 pandemic appeared as influencing factors of both anxiety and depression.

### Conclusion

Our findings suggest that COVID-19 has a substantial effect on the mental health of hypertensive patients. This study highlights the need to develop early intervention and coping

**Data Availability Statement:** All data files are available from the openICPSR [Inter-university Consortium for Political and Social Research]

public repository database (URL: https://www.openicpsr.org/openicpsr/project/206621/version/V1/view DOI: https://doi.org/10.3886/E206621V1).

**Funding:** The author(s) received no specific funding for this work.

**Competing interests:** The authors have declared that no competing interests exist.

strategies among this population to minimize the negative impact of COVID-19 on their mental health and well-being.

## Introduction

COVID-19 whose first case was identified in Wuhan, Hubei, China in December 2019 became a global pandemic that caused several deaths and profoundly impacted public health, economies, and societal well-being worldwide [1]. The World Health Organization (WHO) declared it a Public Health Emergency of International Concern in January 2020 [2] and it led to 0.5 to 2.9% of global deaths over 5 years (1/2020-12/2024) [3]. In a developing country like Nepal, the major consequence of COVID-19 was an increase in mental health problems such as anxiety and depression [4]. The first COVID-19 case was identified on 23rd January 2020 in Nepal [5] significantly leading to the rise of mental health issues [6]. The WHO reports that the increase in mental health problems like depression and suicide are on the rise globally, in the light of this pandemic [7]. People with underlying conditions are more prone to severe infections [8]. Globally, vulnerable groups, especially chronic disease patients are susceptible to infection and deaths due to COVID-19 [9].

In low and middle-income countries, approximately 75% of the population lives with hypertension [10]. In Nepal, the prevalence of hypertension is in increasing trend, from 29% in 2010 to 32% in 2020 [11]. A study in Asia reports hypertensive patients are more prone to the severity of the COVID-19 virus [12] and a similar study significantly links the severity of hypertension with COVID-19 [13]. The prevalence of anxiety was 31.0% and depression was 34.0% during the COVID-19 pandemic in Nepal [14]. In the aftermath of the outbreak, anxiety and depression are more prominent among patients with hypertension [15]. Likewise, findings in Bangladesh estimate that depression and anxiety were 39.9% and 35.2%, respectively in the population with pre-existing medical conditions during COVID-19 [16]. Another study conducted in Ghana and Nigeria demonstrates the prevalence of depression in the hypertensive population [17]. These findings from low and middle-income countries provide evidence of the urgency of information on the burden of anxiety and depression among patients with hypertension in Nepal.

Mental health-related interventions and health policy reforms are crucial requirements, especially during a pandemic. The unpredictability of the pandemic worsens mental health amongst the population with a chronic medical condition [16]. This study intends to address the gaps in anxiety and depression in Nepal by identifying the prevalence and factors associated with anxiety and depression amongst hypertensive patients due to the COVID-19 pandemic and contribute its efforts to the formation of relevant policies.

## Materials and methods

### Study design and study setting

The study design was quantitative and cross-sectional.

The study area was Kathmandu Metropolitan, Bagmati Province, Nepal. Kathmandu is the most populous and largest city of Bagmati province as well in Nepal as a whole. It is also the capital city of Nepal. This site was selected based on the high prevalence of hypertension in Bagmati Province, as reported in the STEPS Survey, 2019 [18].

## Study population

The study population was the adult population aged 40–59 years who were diagnosed with hypertension as this age group is commonly affected by hypertension, which is a significant risk factor for cardiovascular diseases. The pandemic has had a notable impact on mental health, and studying this specific age group can provide insights into the effects of COVID-19 on individuals with pre-existing health conditions like hypertension.

## Sample size

A sample size of 374 was determined based on the sampling formula $n = z^2pq/E^2$ with a 95% confidence interval, 5% margin of error, and 0.42% prevalence of hypertension in 40–60 years of population. The data was collected from May 5, 2021, to June 6, 2021.

## Sampling technique

A non-probability convenience sampling technique was used for selecting the required sample for this study. Among the total of 32 wards, 2 wards were selected as per convenience. Probability proportional to size sampling was then employed to identify the required sample taking into account the proportion of the population from the selected wards of the metropolitan area.

## Selection criteria

### Inclusion criteria.

- The study includes all individuals aged 40–59 residing in selected wards and diagnosed with hypertension at least one year before the COVID-19 pandemic.

### Exclusion criteria.

- Individuals who were bedridden, not able to communicate, needed special care, and suffered from chronic conditions like cancer, heart failure, and kidney failure.

## Ethical considerations

Ethical approval was taken from the Institutional Review Committee (IRC), Pokhara University (Ref. No. 51/077/078). Permission to conduct the study was taken from ward office-26 of Kathmandu Metropolitan. Written or verbal informed consent was obtained from the participants since all the participants were not able to write. The participants were also fully informed about the nature of the research in their local language. Confidentiality, anonymity, and privacy of information was maintained.

## Research tools and their development

Socio-demographic variables, disease-related characteristics, lifestyle-related parameters, and COVID-19-related characteristics were obtained in the first section of the questionnaires.

In the second section of the questionnaire, Generalized Anxiety Disorder (GAD-7), a self-administered questionnaire was used, a tool to measure anxiety, developed by Robert L. Spitzer et al. The anxiety symptoms were calculated by assessing the GAD-7 tool where the scores "0–4", "5–9", "10–14", and "≥15" indicates minimal, mild, moderate, and severe anxiety respectively. The minimal and mild were merged to indicate the absence of anxiety (≤10)

whereas, moderate and severe were merged to show the presence of anxiety ($\geq$10). Its total score ranges from 0 to 21 with a cut-off point of sensitivity of 89% and specificity of 82% [19].

Lastly, the Patient Health Questionnaire (PHQ-9), a self-administered questionnaire was used, a tool to measure depression, developed by Robert L. Spitzer et al. The depression symptoms were calculated by assessing the PHQ-9 tool where the scores "0–4", "5–9", "10–14", and "$\geq$15" indicates minimal, mild, moderate, and severe depression respectively. The Composite International Diagnostic Interview (CIDI) has validated Nepali PHQ-9 with a cut-off $\geq$10: sensitivity 0.94 and specificity 0.80. The minimal and mild were merged to indicate the absence of a depression ($\leq$10) whereas, moderate and moderately severe were merged to show the presence of depression ($\geq$10). It ranges from 0 to 27 and can be scored from 0 (not at all) to 3 (nearly every day). It has a sensitivity and specificity of 88% [19].

## Operational definition

The dependent variable, ethnicity refers to the caste of the individuals. Ethnicity is categorized as per the "Caste/ Ethnicity Classification of Nepal" in the study. For statistical analysis, it has been further grouped into 2 broader categories, merging "Dalit, Disadvantaged Janjatis, Disadvantaged non-Dalit Terai caste group, Religious Minorities" into one group as "Relatively disadvantaged group" and combining "Relatively advanced Janjatis and Upper Caste Groups" into another group as "Relatively advantaged group".

Access to information about COVID-19 refers to the participants who have access to authentic information about the current situation of COVID-19 from sources like social media, news channels, newspapers, or websites of WHO, CDC, or others.

Stayed in isolation refers to the participants who separated themselves from family and friends due to the risk of COVID-19 transmission in a health facility or their residence during the pandemic.

Stayed in quarantine refers to the participants who have been quarantined in a health facility, hotel, or residence in suspect of carrying COVID-19.

## Pretesting, validity, and reliability

The pretesting of the tools was done on 10% (37) of the sample size (374) to check the appropriateness of the questionnaire. The pretested data was entered and analyzed, and necessary modifications were made in the data collection tool. We made a few small modifications to the question wording based on pre-testing. The modifications were done in consultation with the supervisor. An extensive literature review and consultation were done with the supervisor to ensure validity. The interview schedule was back-translated (English-Nepali-English) for easy understanding.

## Data collection technique

The data collection technique was a face-to-face interview using an interview schedule. After obtaining verbal or written consent, interviews with the qualified population were conducted. Before consent, the study's objective and purpose were clearly described. Confidentiality was also maintained. Data collection was conducted in the Nepali language.

## Data management and analysis

Raw data was cleaned, coded, and entered by using EPI DATA version 3.1, and all the entered data was transferred to Statistical Package for Social Sciences (SPSS version 16) for further analysis, and data analysis was done by using a data analysis plan. Descriptive statistics were

calculated for socio-demographic variables, disease-related characteristics, lifestyle-related parameters, and COVID-19-related characteristics as well as anxiety and depression. The chi-square test was used to identify the statistical significance between dependent and independent variables. Logistic regression was applied to identify the factors associated with anxiety and depression.

## Results

Socio-demographic characteristics. Out of a total of 374 respondents, two-thirds of them were between the ages of 50–59 years old (75.1%) and male (75.7%). Most of the participants were from relatively advantaged ethnic groups (91.2%). Almost half of the respondents had completed their secondary education (43.9%). The practice of self-owned business (42.2%) was higher compared to other occupations and the maximum monthly income was noted to be up to Rs.20, 000 (33.4%). Most of the participants lived in a nuclear family (68.4%). (Table 1).

**Table 1. Socio-demographic characteristics (n = 374).**

| Variables | Frequency (n) | Percentage (%) |
|---|---|---|
| **Age (years)** | | |
| 40–49 | 93 | 24.9 |
| 50–59 | 281 | 75.1 |
| Mean Age = 52.71 years, S.D. = 3.996, Minimum = 43 years, Maximum = 59 years | | |
| **Sex** | | |
| Male | 283 | 75.7 |
| Female | 91 | 24.3 |
| **Ethnicity** | | |
| Relatively disadvantaged groups | 33 | 8.8 |
| Relatively advantaged groups | 341 | 91.2 |
| **Education level** | | |
| No formal education | 15 | 4.0 |
| Basic education | 28 | 7.5 |
| Secondary education | 164 | 43.9 |
| Undergraduate | 34 | 9.1 |
| Graduate/above | 133 | 35.6 |
| **Occupation** | | |
| Service: Government/Private | 142 | 38.0 |
| Agriculture | 14 | 3.7 |
| Self-owned business | 158 | 42.2 |
| Daily wage labor | 2 | 0.5 |
| Foreign service | 1 | 0.3 |
| Housewife | 57 | 15.2 |
| **Monthly income** | | |
| Up to 10,000 | 99 | 26.5 |
| Up to 20,000 | 125 | 33.4 |
| Up to 30,000 | 122 | 32.6 |
| Up to 40,000 | 28 | 7.5 |
| **Types of family** | | |
| Nuclear | 256 | 68.4 |
| Joint/ Extended | 118 | 31.6 |

**Table 2. Anxiety and depression symptoms (n = 374).**

| Variables | Frequency(n) | Percentage (%) |
|---|---|---|
| **Anxiety** | | |
| Minimal | 38 | 10.2 |
| Mild | 232 | 62.0 |
| Moderate | 103 | 27.5 |
| Severe | 1 | 0.3 |
| **Depression** | | |
| Minimal | 39 | 10.4 |
| Mild | 244 | 65.2 |
| Moderate | 91 | 24.3 |

Anxiety and Depression symptoms. Among the total participants, 27.8% of respondents were suffering from different forms of anxiety symptoms while 24.3% of participants were suffering from different forms of depression symptoms (Table 2).

Association of socio-demographic characteristics with anxiety and depression. None of the variables were found statistically significant with both anxiety and depression in Pearson's chi-square test. (Table 3).

Association between disease's related characteristics and anxiety and depression. The family members with hypertension were significantly associated with anxiety (p<0.005) while none of the variables were found statistically significant with depression in Pearson's chi-square test. (Table 4).

Association of lifestyle-related parameters with anxiety and depression. Smoking/tobacco consumption was highly significant (p < 0.001) with both anxiety and depression while moderate level physical activity and alcohol consumption were highly significant (p < 0.001) with anxiety only. (Table 5).

Association of COVID-19-related characteristics with anxiety and depression. All variables-stayed in quarantine, COVID-19 test result, stayed in isolation, history of COVID-19

**Table 3. Association of sociodemographic characteristics with anxiety and depression.**

| Variables | Anxiety Symptoms | | Chi-square | p-value | Depression Symptoms | | Chi-square | p-value |
|---|---|---|---|---|---|---|---|---|
| | Yes (104; 27.8%) | No (270; 72.2%) | | | Yes (91; 24.3%) | No (283; 75.7%) | | |
| **Age** | | | | | | | | |
| 40–49 years | 29(31.2) | 64(68.8) | 0.702 | 0.402 | 24(25.8) | 69(74.2) | 0.146 | 0.702 |
| 50–59 years | 75(26.7) | 206(73.3) | | | 67(23.8) | 214(76.2) | | |
| **Sex** | | | | | | | | |
| Male | 84(29.7) | 199(70.3) | 2.036 | 0.154 | 72(25.4) | 211(74.6) | 0.779 | 0.378 |
| Female | 20(22.0) | 71(78.0) | | | 19(20.9) | 72(79.1) | | |
| **Ethnicity** | | | | | | | | |
| Relatively disadvantaged groups | 9(27.3) | 24(72.7) | 0.005 | 0.943 | 7(21.2) | 26(78.8) | 0.191 | 0.662 |
| Relatively advantaged groups | 95(27.9) | 246(72.1) | | | 84(24.6) | 257(75.4) | | |
| **Education level** | | | | | | | | |
| No formal education | 100(27.9) | 259(72.1) | 0.010 | 0.920 | 5(33.3) | 10(66.7) | 0.688 | 0.407 |
| Formal education | 4(26.7) | 11(73.3) | | | 86(24.0) | 243(76.0) | | |
| **Types of family** | | | | | | | | |
| Nuclear | 68(26.6) | 188(73.4) | 0.626 | 0.429 | 58(22.7) | 198(77.3) | 1.237 | 0.266 |
| Joint/Extended | 36(30.5) | 82(69.5) | | | 33(28.0) | 85(72.0) | | |

**Table 4. Association of diseases related characteristics with anxiety and depression.**

| Variables | Anxiety Symptoms | | Chi-square | p-value | Depression Symptoms | | Chi-square | p-value |
|---|---|---|---|---|---|---|---|---|
| | Yes (104; 27.8%) | No (270; 72.2%) | | | Yes (91; 24.3%) | No (283; 75.7%) | | |
| **Duration of diagnosis** | | | | | | | | |
| 36–47 months | 37(31.1) | 82(68.9) | 0.938 | 0.333 | 32(26.9) | 87(73.1) | 0.621 | 0.431 |
| 48–59 months | 67(26.3) | 188(73.7) | | | 59(23.1) | 196(76.9) | | |
| **Another disease along with hypertension** | | | | | | | | |
| Yes | 12(28.6) | 30(71.4) | 0.014 | 0.907 | 12(28.6) | 30(71.4) | 0.462 | 0.497 |
| No | 92(27.8) | 240(72.3) | | | 79(23.8) | 253(76.2) | | |
| **Family members with hypertension** | | | | | | | | |
| Yes | 31(40.8) | 45(59.2) | 8.008 | 0.005* | 23(30.3) | 53(69.7) | 1.823 | 0.177 |
| No | 73(24.5) | 225(75.5) | | | 68(22.8) | 230(77.2) | | |

(*statistically significant at p <0.05)

**Table 5. Association of lifestyle-related parameters with anxiety and depression.**

| Variables | Anxiety Symptoms | | Chi-square | p-value | Depression Symptoms | | Chi-square | p-value |
|---|---|---|---|---|---|---|---|---|
| | Yes (104; 27.8%) | No (270; 72.2%) | | | Yes (91; 24.3%) | No (283; 75.7%) | | |
| **Smoking/Tobacco Consumption** | | | | | | | | |
| Yes | 49(53.3) | 43(46.7) | 39.377 | <0.001* | 36(39.1) | 56(60.9) | 14.514 | <0.001* |
| No | 55(19.5) | 227(80.5) | | | 55(19.5) | 227(80.5) | | |
| **Alcohol Consumption** | | | | | | | | |
| Yes | 74(36.1) | 131(63.9) | 15.531 | <0.001* | 54(26.3) | 151(73.7) | 0.995 | 0.318 |
| No | 30(17.8) | 139(82.2) | | | 37(21.9) | 132(78.1) | | |
| **Moderate Level Physical Activity** | | | | | | | | |
| Yes | 76(24.1) | 240(75.9) | 14.326 | <0.001* | 72(22.8) | 244(77.2 | 2.648 | 0.104 |
| No | 28(48.3) | 30(51.7) | | | 19(32.8) | 39(67.2) | | |

(*statistically significant at p <0.05)

positive in family, history of death due to COVID-19 in family, visiting a hospital during COVID-19 pandemic were significantly associated with both anxiety and depression in Pearson's chi squire test (p<0.05). (Table 6).

Bivariate logistic regression for factors having significant association with anxiety and depression. Smoking/tobacco consumption (OR: 4.703, CI.95%: 2.840–7.789); (OR: 2.653, CI.95%: 1.590–4.427), stayed in quarantine (OR: 13.500, C.I. 95%: 2.397–76.037); (OR: 7.500, C.I. 95%: 1.382–40.690), COVID-19 test result (OR: 2.330, C.I. 95%: 1.316–4.125); (OR: 2.114, C.I. 95%: 1.183–3.779), history of COVID-19 positive in family (OR: 2.033, C.I. 95%: 1.249–3.310); (OR: 2.006, C.I. 95%: 1.203–3.347), history of death due to COVID-19 in family (OR: 3.558, C.I. 95%: 1.719–7.365); (OR: 2.906, C.I. 95%: 1.399–6.036), visiting a hospital during the COVID-19 pandemic (OR: 2.324, C.I. 95%: 1.378–3.919); (OR: 2.071, C.I. 95%: 1.207–3.554) appeared as powerful influencing factors of both anxiety and depression respectively. (Table 7).

## Discussion

The different study variables included in the study were Socio-demographic characteristics, Disease-related characteristics, lifestyle-related parameters, and COVID-19-related

**Table 6. Association of COVID-19-related characteristics with anxiety and depression.**

| Variables | Anxiety Symptoms | | Chi-square | p-value | Depression Symptoms | | Chi-square | p-value |
|---|---|---|---|---|---|---|---|---|
| | Yes (104; 27.8%) | No (270; 72.2%) | | | Yes (91; 24.3%) | No (283; 75.7%) | | |
| **Stayed in quarantine** | | | | | | | | |
| Yes | 9(81.8) | 2(18.2) | 11.055 | 0.001* | 9(81.8) | 2(18.2) | 6.435 | 0.011* |
| No | 8(25.0) | 24(75.0) | | | 12(37.5) | 20(62.5) | | |
| **COVID-19 test result** | | | | | | | | |
| Yes | 66(41.3) | 94(58.8) | 8.633 | 0.003* | 60(37.5) | 100(62.5) | 6.518 | 0.011* |
| No | 22(23.2) | 73(76.8) | | | 21(22.1) | 74(77.9) | | |
| **Stayed in isolation** | | | | | | | | |
| Yes | 69(42.6) | 93(57.4) | 7.113 | 0.008* | 63(38.9) | 99(61.1) | 8.894 | 0.003* |
| No | 13(76.5) | 4(23.5) | | | 13(76.5) | 4(23.5) | | |
| **History of COVID-19 positive in family** | | | | | | | | |
| Yes | 74(33.3) | 148(66.7) | 8.309 | 0.004* | 65(29.3) | 157(70.7) | 7.263 | 0.007* |
| No | 30(19.7) | 122(80.3) | | | 26(17.1) | 126(82.9) | | |
| **History of death due to COVID-19 in family** | | | | | | | | |
| Yes | 18(54.5) | 15(45.5) | 12.889 | <0.001* | 15(45.5) | 18(54.5) | 8.771 | 0.003* |
| No | 86(25.2) | 255(74.8) | | | 76(22.3) | 265(77.7) | | |
| **Visiting a hospital during the COVID-19 pandemic** | | | | | | | | |
| Yes | 33(42.3) | 45(57.7) | 10.322 | 0.001* | 28(35.9) | 50(64.1) | 7.161 | 0.007* |
| No | 71(24.0) | 225(76.0) | | | 63(21.3) | 233(78.7) | | |

(*statistically significant at p <0.05)

characteristics. The results of bivariate logistic regression analysis showed that smoking/tobacco consumption (OR: 4.703, CI.95%: 2.840–7.789); (OR: 2.653, CI.95%: 1.590–4.427), staying in quarantine (OR: 13.500, C.I. 95%: 2.397–76.037), positive COVID-19 test result (OR: 2.330, C.I. 95%: 1.316–4.125); (OR: 2.114, C.I. 95%: 1.183–3.779), history of COVID-19 positive in family (OR: 2.033, C.I. 95%: 1.249–3.310); (OR: 2.006, C.I. 95%: 1.203–3.347), history of death due to COVID-19 in family (OR: 3.558, C.I. 95%: 1.719–7.365); (OR: 2.906, C.I. 95%: 1.399–6.036), and visiting a hospital during the COVID-19 pandemic (OR: 2.324, C.I. 95%: 1.378–3.919); (OR: 2.071, C.I. 95%: 1.207–3.554) appeared to be powerful influencing factors of both anxiety and depression respectively.

In this study, 27.8% of the respondents reported anxiety and 24.3% reported depression during the COVID-19 pandemic. A previous study conducted in India also showed similar results i.e. 28% had anxiety and 25.1% had depression [20]. The findings from a study conducted in Hong Kong mentioned the deterioration in mental health during the COVID-19 pandemic to be 25.4% [21]. The consistent findings show that a public health emergency like the COVID-19 pandemic has substantially caused mental health issues among vulnerable groups like hypertensive patients. This might be due to the constant fear and uncertainty of communicable diseases like COVID-19 which has greater negative health impacts on vulnerable groups like hypertensive patients. Additionally, disruptions in healthcare services can lead to delays in managing hypertension. However, a study conducted among hypertensive patients in Ethiopia found a slightly higher prevalence rate of anxiety (32%) but a significantly lower prevalence rate of depression (5.73%) [22]. The reason for this discrepancy could be that the study in Ethiopia wasn't conducted during the COVID-19 pandemic.

This study showed that smokers/tobacco consumers are 4.703 times more likely to develop anxiety and 2.653 times more likely to develop depression. A study conducted in Ethiopia also

**Table 7. Bivariate logistic regression for factors having significant association with anxiety and depression.**

| Variables | Anxiety | | Depression | |
|---|---|---|---|---|
| | UOR (95% CI) | p-value | UOR (95% CI) | p-value |
| **Family members with hypertension** | | | | |
| Yes | 2.123(1.252–3.601)* | 0.005* | 1.468(0.839–2.568) | 0.179 |
| No | 1 | | 1 | |
| **Smoking/Tobacco Consumption** | | | | |
| Yes | 4.703(2.840–7.789)* | <0.001* | 2.653(1.590–4.427)* | <0.001* |
| No | 1 | | 1 | |
| **Alcohol Consumption** | | | | |
| Yes | 2.617(1.609–7.4.258)* | <0.001* | 1.276(0.790–2.060) | 0.319 |
| No | 1 | | 1 | |
| **Moderate Level Physical Activity** | | | | |
| Yes | 2.947(1.657–5.243) | <0.001* | 1.651(0.899–3.033) | 0.106 |
| No | 1 | | 1 | |
| **Stayed in quarantine** | | | | |
| Yes | 13.500(2.397–76.037)* | 0.003* | 7.500(1.382–40.690) | 0.020* |
| No | 1 | | 1 | |
| **COVID-19 test result** | | | | |
| Yes | 2.330(1.316–4.125)* | 0.004* | 2.114(1.183–3.779) | 0.012* |
| No | 1 | | 1 | |
| **Stayed in isolation** | | | | |
| Yes | 0.228(0.071–0.730) | 0.013 | 0.196(0.061–0.627) | 0.006 |
| No | 1 | | 1 | |
| **History of COVID-19 positive in family** | | | | |
| Yes | 2.033(1.249–3.310)* | 0.004* | 2.006(1.203–3.347)* | 0.008* |
| No | 1 | | 1 | |
| **History of death due to COVID-19 in family** | | | | |
| Yes | 3.558(1.719–7.365)* | 0.001* | 2.906(1.399–6.036)* | 0.004* |
| No | 1 | | 1 | |
| **Visiting a hospital during the COVID-19 pandemic** | | | | |
| Yes | 2.324(1.378–3.919)* | 0.002* | 2.071(1.207–3.554)* | 0.008* |
| No | 1 | | 1 | |

(*statistically significant at p <0.05)

suggested a higher prevalence of anxiety among those who smoke/consume tobacco [22]. Similarly, a study conducted in urban Nepal showed that smoking is 5 times more likely a factor for depression [23]. A study conducted in Turkey [24] suggested that respondents who tend to smoke have a moderate level of depression. Smoking/ tobacco consumption could be significantly associated with anxiety and depression as people may use smoking/ tobacco consumption as a coping mechanism but over time it can cause more harm. Furthermore, anxiety and depression are made worse by the inability to access healthcare facilities and the disruption of support for quitting smoking during the pandemic.

The findings in this study reported that staying in quarantine was 13.5 times more likely to be a factor of anxiety and 7.5 times more likely to be a factor of depression. A study conducted after the outbreak of coronavirus disease in 2019 suggested that quarantine during disease outbreaks can lead to frustration and boredom which could lead to issues like anxiety and

depression. [25] This may be due to the worry of catching the virus and managing their health as well as a lack of social connection and support during the pandemic.

This study revealed that both being tested positive for COVID-19 and family members tested positive for COVID-19 had a significant association with anxiety and depression. Consistent with this finding, a study conducted in the Republic of Ireland reported that respondents whose family members had screened and confirmed the presence of COVID-19 had a higher level of anxiety and depression. [26] Family members' health becomes a major concern, causing elevated anxiety and depression over their state and likely outcomes. The personal health concerns of being diagnosed with COVID-19, increased vulnerability, and concerns about the virus's spread and contagiousness are exacerbated by the need to take precautions.

This study reported that respondents who had a family history of death due to COVID-19 were 3.558 times more likely to develop anxiety and 2.906 times more likely to develop depression during the COVID-19 pandemic. Similarly, a study conducted among American adults [27] also reported that respondents who experienced the death of a loved one due to the pandemic are more likely to develop anxiety and depression. The isolation and lack of support, complicated grief with not having the chance to do a proper last farewell to the loved one might have led to this vulnerability.

This study revealed that the respondents visiting a hospital during the COVID-19 pandemic were 2.324 times more likely to develop anxiety and 2.071 times more likely to develop depression. The reason for this may be due to the fear of infection, COVID-19-related measures, limited social support, uncertainty and disruption of care, and heightened health concerns.

## Strength and limitation

This study represents a pioneering effort within the Kathmandu Metropolitan area to evaluate the mental health impact of the COVID-19 pandemic on hypertensive patients. It aims to ascertain the prevalence of anxiety and depression and identify associated factors during this unprecedented health crisis.

A limitation of this research is its focus on hypertensive adults aged 40–59 years, potentially limiting the representation of all hypertensive individuals. Excluding bedridden/severe chronic patients may reduce the findings' applicability. Future studies should broaden demographics for better generalizability. The cross-sectional design hinders establishing causality between factors and mental health outcomes. Self-reported data and convenience sampling may introduce potential bias. While offering patient insights, the study overlooks healthcare provider perspectives crucial for comprehensive pandemic mental health understanding.

## Conclusion

The study highlights that key factors like smoking/ tobacco consumption, staying in quarantine, being tested positive for COVID-19, having a history of COVID-19 positive in the family, death due to COVID-19 in the family, and hospital visits during the COVID-19 pandemic are associated with symptoms of anxiety and depression among hypertensive patients during COVID-19 pandemic.

The findings obtained from this study conclude that hypertensive patients who are prone to mental health problems like anxiety and depression should be identified and screened and further provided with counseling and awareness to reduce the prevalence of anxiety and depression. It is further implied that those hypertensive patients who have stayed or are staying in isolation and quarantine should be continuously monitored and observed to minimize their association with anxiety and depression. Moreover, hypertensive patients who themselves have tested positive for COVID-19, have a family history of being positive for COVID-19, and

suffered the loss of a family member due to the COVID-19 pandemic should be reached with special care and attention including motivational sessions from health workers.

## Acknowledgments

The authors wish to thank all the respondents for providing their consent and time to participate in the research and all the helping hands who helped either directly or indirectly during the conduction of the research.

## Author Contributions

**Conceptualization:** Dilasha K. C., Hari Prasad Kaphle.

**Data curation:** Dilasha K. C., Hari Prasad Kaphle, Durga Shrestha, Nirmala Neupane.

**Formal analysis:** Dilasha K. C., Hari Prasad Kaphle, Durga Shrestha, Nirmala Neupane.

**Investigation:** Dilasha K. C., Hari Prasad Kaphle.

**Methodology:** Dilasha K. C., Hari Prasad Kaphle.

**Project administration:** Dilasha K. C., Hari Prasad Kaphle, Durga Shrestha, Nirmala Neupane.

**Resources:** Dilasha K. C., Hari Prasad Kaphle, Durga Shrestha, Nirmala Neupane.

**Software:** Dilasha K. C., Durga Shrestha.

**Supervision:** Hari Prasad Kaphle, Nirmala Neupane.

**Validation:** Dilasha K. C., Hari Prasad Kaphle, Durga Shrestha, Nirmala Neupane.

**Visualization:** Dilasha K. C., Hari Prasad Kaphle, Durga Shrestha, Nirmala Neupane.

**Writing – original draft:** Dilasha K. C., Durga Shrestha.

**Writing – review & editing:** Dilasha K. C., Hari Prasad Kaphle, Durga Shrestha, Nirmala Neupane.

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
