## [Decision Letter · Decision Letter 0]

12 Apr 2024

PONE-D-23-25764Anxiety and Depression among Hypertensive Patients during the COVID-19 Pandemic: A Cross-Sectional Study from Kathmandu MetropolitanPLOS ONE

Dear Dr. Shrestha,

Thank you for submitting your manuscript to PLOS ONE. After careful consideration, we feel that it has merit but does not fully meet PLOS ONE’s publication criteria as it currently stands. Therefore, we invite you to submit a revised version of the manuscript that addresses the points raised during the review process.

It is an interesting study from LMIC during covid pandemic and address important illness of depression

We look forward to receiving your revised manuscript.

Kind regards,

Aysha Almas, MBBS, FCPS, MSc

Academic Editor

PLOS ONE

4. We are unable to open your Supporting Information file [S1 file.sav]. Please kindly revise as necessary and re-upload.

Academic Editor comments:

Please address the following comments from reviewers

Review

Reviewers' comments:

Reviewer's Responses to Questions

**Comments to the Author**

1. Is the manuscript technically sound, and do the data support the conclusions?

Reviewer #1: Partly

Reviewer #2: Yes

Reviewer #3: Yes

2. Has the statistical analysis been performed appropriately and rigorously? 

Reviewer #1: No

Reviewer #2: Yes

Reviewer #3: Yes

3. Have the authors made all data underlying the findings in their manuscript fully available?

Reviewer #1: Yes

Reviewer #2: Yes

Reviewer #3: Yes

4. Is the manuscript presented in an intelligible fashion and written in standard English?

Reviewer #1: Yes

Reviewer #2: No

Reviewer #3: No

5. Review Comments to the Author

Reviewer #1: The authors present findings from their cross-sectional study regarding anxiety and depression in hypertensive patients in Kathmandu during the COVID-19 pandemic. I am not clear on what new knowledge findings of this study add to the available evidence. The introduction section states hypertension as a problem, however, doesn't apprise the reader of anxiety/depression prevalence in the study setting. The inclusion criteria was based on including patients with hypertension only, however given the rise of multimorbidity in low and middle income countries, this is not representative of the general population. Contrary to what authors state in the manuscript about the findings being representative of the people with hypertension, however this may not be the case because of the inclusion/exclusion criteria and use of a convenience sampling technique.

Why did the authors decide to use bivariate logistic regression analysis and not multivariate?

Authors mention as part of limitations that data regarding pre-existing diagnosis of anxiety and/or depression was collected, however this is information would have presented a close approximation of the magnitude of association of COVID-19 pandemic with anxiety and/or depression in this population.

Reviewer #2: Overall

The authors have discussed an important aspect of health in hypertensive patients: Mental well being. The authors however need to make some changes in the manuscript prior to its acceptance. Moreover, there are grammatical mistakes, and some sentences do not make sense due to absence of a comma or semicolon. One example is given below

"The WHO reports that the increase in mental health problems like depression and suicide is on the rise globally [4], in the light of this pandemic."

Instead of 'is' it should be are

Introduction

1. Covid-19 has changed a lot in the past two years. The figures cited in the initial paragraph are from 2020 and do not give a true picture of what is happening currently.

2. The authors write "It is also noted that in the case of Nepal, vulnerable groups, especially chronic disease patients are susceptible to infection and deaths due to COVID19". This is true globally and not just for Nepal

3. The authors have no cited any similar studies or any studies related to prevalence of hypertension or depression in Introduction. The authors fail to convince as to why they have chosen hypertensive patients. As per WHO "Nearly 1 in 4 people in Nepal suffer from hypertension, and less than 5% of hypertension patients have the condition under control" These type of figures should be quoted. In addition studies on Anxiety and Depression among Hypertensive Adults in Nepal should be cited and rationale be made regarding covid-19

Methodology

1. Cite the STEPS survey

2. Why only 40-59….are we only looking at middle-aged adults

3. Good validation, reliability and pretesting

4. No definition of ethnicity in methodology

Results

1. Self-owned business – better word instead of own business

2. Was moderate level physical activity self reported or calculated using a IPAQ questionnaire

3. Too many tables. Table 2 and 3 can be supplementary

4. Table 7 could be better if Models were created

Discussion

5. Better to have a complete summary of results in 1st paragraph

6. Typically people who smoke tend not to be worried about their health risks due to smoking. There is also lack of reference citation

7. Explanations for risk factors behind anxiety are much or less the same. They can be grouped together rather than writing again.

8. Rather than stating different periods as the reason behind differences in finding, the study in Ethiopia didn’t look at covid-19 pandemic

9. Any strengths that can be included in the discussion

Conclusion

10. Rather than stating anxiety and depression, it would be better to state that the risk factors had an association with "symptoms" of anxiety and depression

Reviewer #3: minor grammatical errors are present during the writing of this manuscript especially in the introduction and discussion section. The discussion section must be made more robust. Other comments have been highlighted with comments.

6. PLOS authors have the option to publish the peer review history of their article (what does this mean?). If published, this will include your full peer review and any attached files.

Reviewer #1: No

Reviewer #2: **Yes: **Saad Bin Zafar Mahmood

Reviewer #3: **Yes: **Zeerak Jarrar

---

## [Author Response · Author response to Decision Letter 0]

25 Jun 2024

Journal requirements

 1- When submitting your revision, we need you to address these additional requirements.

Response: Thank you for your valuable feedback. We appreciate your feedback. We have provided our best efforts to meet the journal requirements. 

2-Did you know that depositing data in a repository is associated with up to a 25% citation advantage (https://doi.org/10.1371/journal.pone.0230416)? If you’ve not already done so, consider depositing your raw data in a repository to ensure your work is read, appreciated and cited by the largest possible audience. You’ll also earn an Accessible Data icon on your published paper if you deposit your data in any participating repository (https://plos.org/open-science/open-data/#accessible-data).

Response: Thank you for your feedback and the valuable information.

3- Please include captions for your Supporting Information files at the end of your manuscript, and update any in-text citations to match accordingly. Please see our Supporting Information guidelines for more information: http://journals.plos.org/plosone/s/supporting-information. 

Response: Thank you for your feedback. We have removed the data from Supporting file as it was not able to open and have deposited in an appropriate public repository as suggested.

4- We are unable to open your Supporting Information file [S1 file.sav]. Please kindly revise as necessary and re-upload.

Response: Thank you for your valuable comment. We have revised and re-uploaded it again for your kind perusal. We have deposited in an appropriate public repository. The published data can be accessed through the URL: https://www.openicpsr.org/openicpsr/project/206621/version/V1/view and it’s DOI is https://doi.org/10.3886/E206621V1

Reviewer #1 

A) The authors present findings from their cross-sectional study regarding anxiety and depression in hypertensive patients in Kathmandu during the COVID-19 pandemic. I am not clear on what new knowledge findings of this study add to the available evidence. 

The introduction section states hypertension as a problem, however, doesn't apprise the reader of anxiety/depression prevalence in the study setting.

Response: This study adds valuable data on the prevalence and risk factors of anxiety and depression in hypertensive patients in a specific low-income setting like Nepal during the COVID-19 pandemic. This is crucial for developing tailored interventions and coping strategies that are culturally and socio-economically appropriate in local context. In addition, the study underscores the need for early intervention and the development of coping strategies to mitigate the mental health impacts of the pandemic. This can inform public health policies and healthcare practices in similar low-income settings, emphasizing the importance of mental health support alongside physical health management for hypertensive patients.

Thank you for your comment. Now, we have acknowledged it, further corrected the introduction and focused on the cause of this study. Please see pages 3 and 4 in the revised manuscript.

B) The inclusion criteria was based on including patients with hypertension only, however given the rise of multimorbidity in low and middle income countries, this is not representative of the general population. Contrary to what authors state in the manuscript about the findings being representative of the people with hypertension, however this may not be the case because of the inclusion/exclusion criteria and use of a convenience sampling technique.

Response: We accept your valuable comment. Due to resource constraints, and as a practical option we particularly focused on hypertension in Nepal in our study. Now we have acknowledged it in our study limitations. Please see page no 22 and 23 in the revised manuscript.

C) Why did the authors decide to use bivariate logistic regression analysis and not multivariate?

Response: Thank you for your query. The bivariate logistic regression was used due to the limited sample size. Moreover, this cross-sectional study is a preliminary step before conducting subsequent studies followed by multivariate logistic regression analysis. 

D) Authors mention as part of limitations that data regarding a pre-existing diagnosis of anxiety and/or depression was collected, however this is information would have presented a close approximation of the magnitude of association of COVID-19 pandemic with anxiety and/or depression in this population.

Response: Thank you for your valuable comment. As a researcher, we have tried our best to incorporate information on anxiety and depression. However, we acknowledge the limitations of an absence of pre-existing data on anxiety and depression, mostly due to resource constraints in reaching pre-diagnosed patients in the pandemic, and have further elaborated on it in our study limitations. Please see page no. 22 and 23 in the revised manuscript.

Reviwer#2

A) The authors have discussed an important aspect of health in hypertensive patients: Mental well being. The authors however need to make some changes in the manuscript prior to its acceptance. Moreover, there are grammatical mistakes, and some sentences do not make sense due to absence of a comma or semicolon. One example is given below

"The WHO reports that the increase in mental health problems like depression and suicide is on the rise globally [4], in the light of this pandemic."

Instead of 'is' it should be are

Response: Thank you for your valuable comment. Now it has been thoroughly revised and addressed in the revised manuscript. Please see page no. 3 and 4 in the revised manuscript. 

B) Introduction

1. Covid-19 has changed a lot in the past two years. The figures cited in the initial paragraph are from 2020 and do not give a true picture of what is happening currently.

Response: Thank you for your valuable comment. We have further solidified the evidence related to COVID-19. We have removed the old figures from 2020 in the revised manuscript. Please see page no 3 and 4 in the revised manuscript. 

2. The authors write "It is also noted that in the case of Nepal, vulnerable groups, especially chronic disease patients are susceptible to infection and deaths due to COVID19". This is true globally and not just for Nepal

Response: We acknowledge your valuable comments. Now it is addressed. Please see page no 3 in the revised manuscript. 

3. The authors have no cited any similar studies or any studies related to prevalence of hypertension or depression in Introduction. The authors fail to convince as to why they have chosen hypertensive patients. As per WHO "Nearly 1 in 4 people in Nepal suffer from hypertension, and less than 5% of hypertension patients have the condition under control" These type of figures should be quoted. In addition studies on Anxiety and Depression among Hypertensive Adults in Nepal should be cited and rationale be made regarding covid-19

Response: We accept your comment. It has been addressed in the revised manuscript and rationale has been added to strengthen the point of this study. Please refer to page no. 3, 4 and 5 in the revised manuscript.

C) Methodology

1. Cite the STEPS survey

Response: Now we have added the citation and reference for the sentences. Please see page no 4 in the revised manuscript. 

2. Why only 40-59….are we only looking at middle-aged adults

Response: Thank you for your insightful comment. It has been addressed in the revised manuscript and rationale has been added to strengthen the point of this study. Please refer to page no. 5 in the revised manuscript.

3. Good validation, reliability and pretesting

Response: Thank you for your valuable comment. 

4. No definition of ethnicity in methodology

Response: Thank you for your valuable comment. Now it is addressed. Please see page no 7 in the revised manuscript. 

D) Results

1. Self-owned business – better word instead of own business

Response: Thank you for your valuable suggestion. Now it is addressed. Please see page no 9 and 10 in the revised manuscript. 

2. Was moderate level physical activity self-reported or calculated using a IPAQ questionnaire

Response: We acknowledge your comment. The moderate level of physical activity was self-reported by the respondents in this study. 

3. Too many tables. Table 2 and 3 can be supplementary

Response: Thank you for your valuable comment. However, we think Table 2 is significant as it provides the prevalence rates of anxiety and depression among hypertensive patients during the COVID-19 pandemic in Kathmandu Metropolitan, Nepal. It is crucial because it quantifies the extent of mental health issues among the study population, which is essential for understanding the impact of the pandemic on this group. The data can inform healthcare providers and policymakers about the need for targeted mental health interventions.

Similarly, Table 3 presents the association of sociodemographic characteristics with anxiety and depression among hypertensive patients during the COVID-19 pandemic which is one of the main objective of this study so, although it is long but we believe it is reasonable.

4. Table 7 could be better if Models were created

Response: We acknowledge your valuable feedback. Unfortunately, due to limited sample size we are currently unable to generate these models.

E) Discussion

5. Better to have a complete summary of results in 1st paragraph

Response: Thank you for your valuable comment. Now, we have acknowledged it. Please see page no. 20 in the revised manuscript. 

6. Typically people who smoke tend not to be worried about their health risks due to smoking. There is also lack of reference citation

Response: Thank you for your valuable comment. We have removed the statement. 

7. Explanations for risk factors behind anxiety are much or less the same. They can be grouped together rather than writing again.

Response: Thank you for your valuable comment. We have acknowledged and revised it. Please visit page no. 19, 20, 21, 22 in the revised manuscript.

8. Rather than stating different periods as the reason behind differences in finding, the study in Ethiopia didn’t look at covid-19 pandemic

Response: Thank you for your insightful comment. We have acknowledged and revised it. Please visit page no. 20, 21 in the revised manuscript.

9. Any strengths that can be included in the discussion

Response: Thank you for your valuable comment. We have acknowledged and revised it. Please visit page no. 22 in the revised manuscript.

F) Conclusion

10. Rather than stating anxiety and depression, it would be better to state that the risk factors had an association with "symptoms" of anxiety and depression.

Response: Thank you for your valuable comment. We have acknowledged and revised it. Please visit page no. 23 in the revised manuscript.

Reviewer#3

A) minor grammatical errors are present during the writing of this manuscript especially in the introduction and discussion section. The discussion section must be made more robust. Other comments have been highlighted with comments.

Response: Thank you for your valuable comment. The correction for grammatical errors has been addressed in the introduction and discussion sections and other highlighted comments have also been addressed.

---

## [Editor Report · Decision Letter 1]

17 Jul 2024

Anxiety and depression among hypertensive patients during the COVID-19 pandemic: A cross-sectional study from Kathmandu Metropolitan, Nepal

PONE-D-23-25764R1

Dear Dr. Shrestha

We’re pleased to inform you that your manuscript has been judged scientifically suitable for publication and will be formally accepted for publication once it meets all outstanding technical requirements.

Kind regards,

Aysha Almas, MBBS, FCPS, MSc

Academic Editor

PLOS ONE
---

## [Editor Report · Acceptance letter]

22 Jul 2024

PONE-D-23-25764R1 

PLOS ONE

Dear Dr. Shrestha, 

I'm pleased to inform you that your manuscript has been deemed suitable for publication in PLOS ONE. Congratulations! Your manuscript is now being handed over to our production team.

Kind regards, 

on behalf of

Dr. Aysha Almas 

Academic Editor

PLOS ONE